# Perinatal Femoral Fracture: A Ten-Year Observational Case Series Study

**DOI:** 10.3390/children7100156

**Published:** 2020-10-01

**Authors:** Vito Pavone, Andrea Vescio, Marco Montemagno, Claudia de Cristo, Ludovico Lucenti, Piero Pavone, Gianluca Testa

**Affiliations:** 1Department of General Surgery and Medical Surgical Specialties, Section of Orthopaedics and Traumatology, University Hospital Policlinico-Vittorio Emanuele, University of Catania, 95123 Catania, Italy; andreavescio88@gmail.com (A.V.); m-acor@hotmail.it (M.M.); gianpavel@hotmail.com (G.T.); 2Dipartimento Ortopedia e Traumatologia-Ospedale Santa Maria Goretti, 04100 Latina, Italy; decristo.claudia@gmail.com; 3Dipartimento Ortopedia e Traumatologia-Policlinico Casilino, 00169 Roma, Italy; ludovico.lucenti@gmail.com; 4Department of Clinical and Experimental Medicine, Section of Pediatrics and Child Neuropsychiatry, University of Catania, 95123 Catania, Italy; ppavone@unict.it

**Keywords:** birth-related fractures, perinatal femoral fractures, delivery trauma, skin traction, Bryant’s traction

## Abstract

Background: perinatal femoral fractures (PFF) are relative rare birth-related fractures. Among treatment options, Bryant traction reported satisfactory outcomes in PFF of children under 3 years of age. The aim of this study is to assess the risk factors, diagnosis, management, and outcome in the 10-year multicentric experiences of all newborns treated for PFF in Catania city hospitals. Methods: 15,628 children, hospitalized in four neonatal units, were retrospectively reviewed. The following data were collected: gender, birth weight, gestational age, presentation, mode delivery, and fracture type according to AO Pediatric Comprehensive Classification of Long Bone Fractures (PCCF). In each case, diagnosis was achieved after the clinical examination and X-Ray exam. Each patient underwent Bryant’s skin traction of the affected limb, and was clinically followed for at least two years. Results: eight newborns were included in the study (five males). The average birth weight was 2.656 kg with a gestational age of 37.5 weeks; 4 cases were preterm birth; 5 patients had a cephalic presentation. According to the AO PCCF classification, three fractures were ranked 32-D/4.1 and five were 32-D/5.1. The entire cohort had an excellent outcome. Conclusions: prematurity, low birth weight, and caesarean section could be PFF risk factors. Bryant’s skin traction is an effective option to achieve an excellent outcome.

## 1. Introduction

Birth fractures are those diagnosed in the first week of life, in the absence of any postnatal trauma [1]. Birth-related fractures result from trauma during delivery. The most common sites are the clavicle and humerus and, rarely, the femur [2,3]. The incidence of these fractures is variable and likely underestimated, ranging between 0.1 and 10.5 per 1000 live births [2,4]. In 1922, Ehrenfest et al. [5] described the first perinatal femoral fracture (PFF), induced by the difficult extraction of a large baby during a cesarean section (CS). Associated risk factors include malpresentation, low birth weight, macrosomia, prematurity, osteogenesis imperfecta, disuse osteoporosis following immobilization, CS [6], difficult extraction (breech presentation), impacted foot in the pelvis, and previous uterine surgery leading to a tight uterine incision. The diagnosis usually occurs immediately after the delivery, but can be delayed in cases with milder symptoms [7]. The most common clinical features are a swollen, tender, warm thigh held motionlessly. Often, clinicians feel a ‘’crack’’ during the delivery. A radiographic assessment confirms the diagnosis, while MRI and ultrasound are often used to make a differential PFF diagnosis [8,9]. Numerous successful PFF treatments have been developed that immobilize the femoral shaft. Spica cast, Pavlik harness, gallows traction, splinting with two tongue depressors, and Bryant’s traction are the most common treatments, and all guarantee an optimal outcome. Since initially proposed in 1873, Bryant’s traction method has been used to successfully treat PFF [10] and developmental hip dysplasia [11] in children under 3 years of age.

This study aims to investigate PFF risk factors, diagnoses, management strategies, and outcomes using 10 years of multicentric data on newborns treated for a PFF in Catania city public hospitals, and compares these results with those from studies published in the last 20 years.

## 2. Materials and Methods

### 2.1. Sample

Between January 2005 and December 2015, 90,243 newborns (~9000/year) were delivered in Catania City’s public hospitals. A computerized retrospective review of 15,628 medical records from hospitalized children was performed, using the code ”820” for femoral fracture. The first screening was selected for patients under one month old. The inclusion criteria were: (1) confirmed diagnosis of displaced PFF; (2) treatment using Bryant’s traction; (3) chronological age under one month at the time of diagnosis; (4) treated per ‘treatment protocol’ described below; (5) follow-up over 6 months; and (5) complete radiographic data. The exclusion criteria were: (1) patients older than one month at the time of injury; (2) open, pathological, syndrome-related, or neurological fracture; (3) follow-up less than 6 months; and (4) incomplete radiographic data.

The collected data included demographic information such as sex, birth weight, gestational age, presentation, mode of delivery, AO Pediatric Comprehensive Classification of Long Bone Fractures (PCCF) [12], other injuries suffered, comorbidities, and treatment modality (Table 1).

### 2.2. Treatment Protocol

In every case, the fracture diagnosis was made after a clinical examination and X-ray. All fractures were treated similarly: Bryant’s skin traction of the affected leg(s) with hip flexion of 90° with the infant’s buttock(s) elevated 1 cm from the cot (weight around 200 g) for 15 days [13,14] (Figure 1).

The infants received clinical and radiological follow-up care after 7, 20, and 60 days until the fracture healed, defined as the bridging of the fracture site by callus, obliteration of the fracture line, and continuity without rotation and axial deformities of the femur. All of the infants were kept in the hospital until the fractures were stabilized. No further immobilization or splinting was needed after the release of the traction. A clinical follow-up was performed every six months for two years and every year until the skeletal maturation. All treatments were performed and data collected and analyzed by the same expert team of surgeons.

## 3. Results

The birth-related femoral fractures group was composed of eight newborns, five (62.5%) male and three (37.5%) female (Figure 2).

The average birth weight was 2.656 Kg (SD: 548.9 g, range: 1.980–3.340 Kg). The cohort’s mean gestational age was 37.5 weeks (SD: 2.61 weeks, range: 34–42 weeks), one infant (12.5%) had a pre-term delivery at 34 weeks and one was born at 42 weeks. Four cases (50%) were preterm births (<37 weeks), three were appropriate for gestational age (AGA), and one was small for gestational age (SGA). The remaining 50% were full-term (37–42 weeks), one of whom was SGA. No newborn was large for gestational age (LGA).

Five of the eight patients had a cephalic presentation. Three (37.5%) neonates required an emergency CS and two (20%) were delivered by elective CSs. According to the AO PCCF classification, three fractures were ranked as 32-D/4.1 and five were 32-D/5.1; only one child presented with femoral and ipsilateral clavicle fractures. The average traction duration for femoral fractures was 16.3 days.

The mean time from delivery to fracture diagnosis was 1.5 days (range: 1–4 days). Hard bone callus formation was seen between 7 and 10 days from treatment initiation. All of the fractures showed clinical and radiologic healing with abundant callus formation within 3 weeks after removing the traction; no further immobilization was necessary.

No severe complications were observed during hospitalization. Six out of eight patients were lost at the follow-up after 1 year from fracture. By the final follow-up, no deformity, shortening, or other significant complications were noted for any of the patients. Only one case had a minor complication (skin sloughing).

## 4. Discussion

According to our data, the PFF incidence was 0.13/1000 live births per 10 years in Catania, Italy. Of the eight cases, five had a cephalic presentation at birth and three were breech. Five infants were delivered by CS and three vaginally. The mean gestational age was 37.5 weeks, but half of the cases were preterm. Only two cases were SGA and none were LGA. All fractures were diagnosed by X-ray and treated with Bryant’s skin traction with optimal radiological and functional outcomes.

PFF is a rare trauma in neonatology units [14]. A Portuguese study of 7364 newborns found 76 fractures related to birth trauma. The most commonly fractured bone was the clavicle (79%), followed by parietal (7%), humerus (5%), and femur (4%) [3]. Morris et al. [4] reported a PFF incidence of 0.13/1000 live births. Basha et al. [1] reported an incidence of 0.17/1000 live births. The incidence rate calculated from our medical records was slightly higher than, but still comparable to, the literature (0.89/1000 live births). In the past, a CS was considered to reduce the risk of fracture [15]. However, according to some studies [7,16], the considerable number of maneuvers and tractions performed during a CS can result in fractures. Vasa et al. [7] emphasize that CS does not reduce the traumatic morbidity in average-sized infants to zero. Burnes and Van Geem [15], Alexander et al. [17] reported midshaft, distal metaphyseal fractures after CS. In our series, CS (62.5% of our cases) could be a risk factor for femur fracture; similarly, Givon et al. [13] recorded an increased risk of PPF during a CS. The reasons underlying this finding are unclear, it was hypothesized that to increase CSs in the management of delivery represents the most important risk, combined with fetal presentation and the obstetrician’s experience.

Many studies report a relationship between femur fractures and CS with a breech presentation [4,16]. In contrast, in the present study, the most represented presentation was cephalic (62.5%). From the literature on PFFs [1,4,6,13], the mean gestational age was 36.7 weeks (SD: 3.55 weeks) and the mean birth weight was 2.726 Kg (SD: 704.97 g). Interestingly, 4/8 (50%) of our patients were premature compared with 11/34 (32%) in the literature [1,4,6,13]. Preterm birth could be associated with a higher risk of perinatal fracture; around 80% of bone mineralization occurs during the third trimester and, therefore, premature newborns have an increased risk of osteopenia [3,18,19]. Other metabolic bone risk factors associated with this injury include osteogenesis imperfecta and disuse osteoporosis following prolonged immobilization [4,6]. In addition, in our study, no LGA patients were recorded. This, combined with published evidence [6], underlines a dissimilarity with the perinatal clavicle fracture related to fetal macrosomia [20].

In our sample, the femoral fracture site was proximal in five patients (62.5%), and the middle third in three (37.5%). The most common fracture type was spiral and involved the femoral shaft. Very few cases of transphyseal, upper epiphyseal, and lower femoral physeal fractures diagnosed by MRI and ultrasound in the first week of life were reported [8].

Bryant’s skin traction consistently achieved optimal outcomes; no deformities, shortenings, or other complications were noted at the last radiological follow-up. Notably, all known treatment regimens provide satisfactory clinical and radiological results [4]; thus, the choice of treatment is best made according to the experience of the operating clinician. In Givon et al. [13], the authors use a modified Bryant’s traction approach adapted for use in a newborn’s cot. This involves tractioning both legs, with the hips flexed to 90°. Pulleys are attached to infusion stands instead of a specialized bed frame, with 100 mL bags of normal saline serving as weights, such that the infant’s buttocks are elevated 1 cm above the cot. Despite the major drawback of Bryant’s traction requiring unduly long hospitalization [21], Givon et al. [13] concluded that Bryant’s traction for 2–3 weeks in a hospital is a safe and effective method for the treatment of PFF in neonates. The principal alternative treatment option is the Pavlik harness. The Pavlik harness addresses the flexed, abducted, and externally rotated position of the proximal fragment. Furthermore, if the reduction is lost, the fracture is easily reduced by adjusting the straps [4]. Stannard et al. [22] report on 16 PFF cases healed using the Pavlik harness in good alignment with no adverse outcomes or complications. Patients show good healing with callus formation and evidence of union after an average of 4 weeks [6]. In a study of seven cases published by Morris et al. [4], three infants were treated with the Pavlik harness, which was removed after 4 weeks. The Pavlik harness’s main advantages are minimal nursing care and short hospitalization, even if minor complications, such as skin sloughing, do occur.

The limitations of this study include the lack of data recorded about the pregnant mothers’ clinical conditions and the low number of newborn cases, both of which prevented a more exhaustive retrospective study. Future studies should include more cases of pathological fractures in newborns, such as osteogenesis imperfecta or metabolic disease, to provide links to other risk factors and enable better differential diagnoses. Another limitation was the lack of detailed data on the delivery process, such as the obstetrician’s techniques and experience, as are reported in other studies [1].

## 5. Conclusions

PFFs are very rare. The data analyzed in the present study confirmed that prematurity, low birth weight, and CS seem to be important predisposing factors. Despite the small sample size, due to the injury’s paucity, our findings support the use of Bryant’s skin traction to achieve optimal patient outcomes, as previously demonstrated in the literature.

## Figures and Tables

**Figure 1 children-07-00156-f001:**
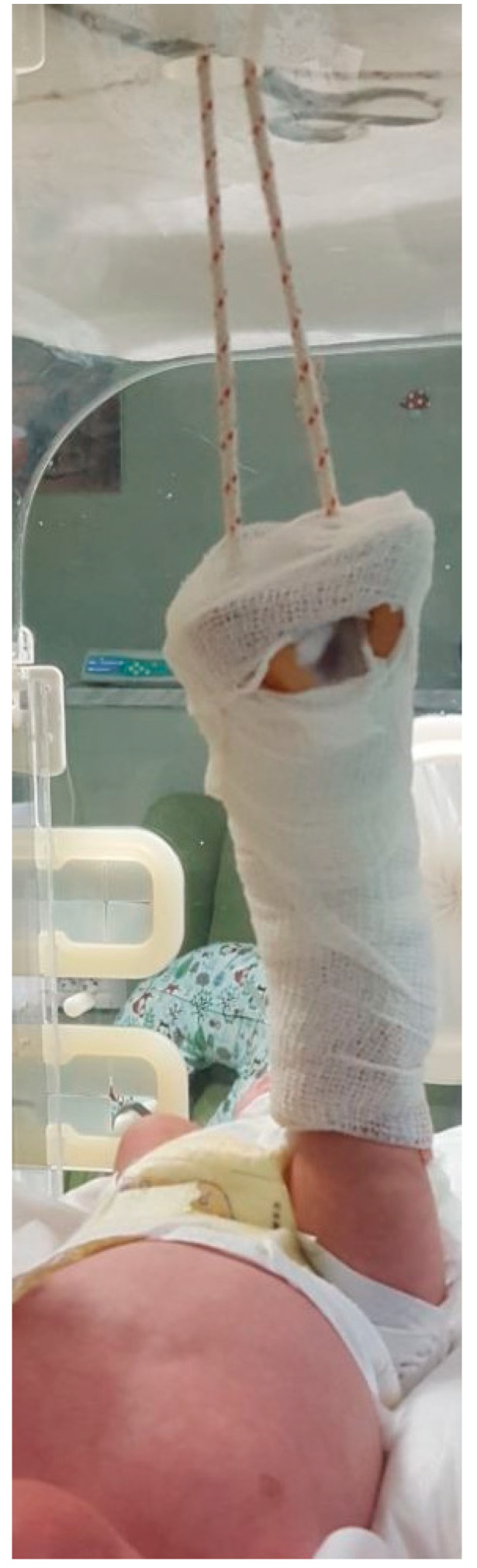
Newborn, 4 days old with perinatal femoral fractures treated with Bryant’s skin traction.

**Figure 2 children-07-00156-f002:**
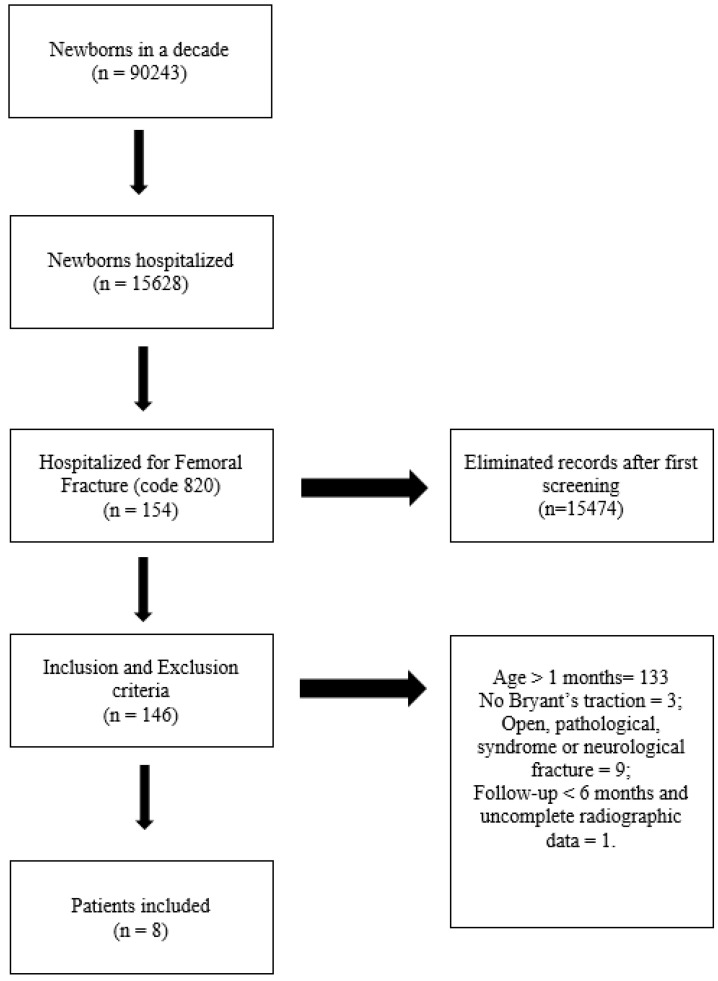
Sample selection scheme.

**Table 1 children-07-00156-t001:** Data on newborns with a perinatal femoral fracture collected from January 2005–December 2015.

Pz	Sex	Birth Weight (kg)	Gestational Age (Weeks)	Presentation	Delivery Mode	Fractures (AO PCCF)	Time to Diagnosis (Days)
**G.P.**	Male	2.050	34	Breech	Cesarean	32-D/5.1	1
**S.M.**	Male	2.580	36	Cephalic	Vaginal	32-D/5.1	1
**E.M.**	Male	3.200	40	Cephalic	Cesarean	32-D/5.1	1
**G.C.**	Female	2.160	36	Cephalic	Cesarean	32-D/4.1	1
**M.P.**	Female	1.980	39	Cephalic	Vaginal	32-D/5.1	1
**F.C.**	Male	3.340	42	Breech	Cesarean	32-D/4.1	4
**L.C.**	Male	3.160	37	Cephalic	Vaginal	32-D/4.1	2
**A.P.**	Female	2.780	36	Breech	Cesarean	32-D/5.1	1

pz = patients; AO PCCF = Pediatric Comprehensive Classification of Long Bone Fractures.

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
