# Peer review of "Perinatal Femoral Fracture: A Ten-Year Observational Case Series Study"

_children, 2020, doi:10.3390/children7100156_

Round 1
Reviewer 1 Report
In their article, Pavone and co-workers aimed to assess the risk factors, diagnosis, management, and outcome in 10 years multicentric experience of all newborns treated for perinatal femoral fracture PFF. Among 15628 children, hospitalized in 4 neonatal units, 8 newborns were included in the study. Prematurity, low birth weight and caesarean section were suspected as PFF risk factors and Bryant’s skin traction was found as an effective option to achieve the excellent outcomes.
The paper is of interest for the research question and the low incidence of PFF. References are adequate and its readability is good. Strengths and limitations are listed by the Authors. Two points require attention:
- Repetitive information in table and text should be avoided.
- Due to the very small size and subsequent limitation to perform the adequate statistical analysis, those listed as ‘risk factors’ require redefinition.
Author Response
In their article, Pavone and co-workers aimed to assess the risk factors, diagnosis, management, and outcome in 10 years multicentric experience of all newborns treated for perinatal femoral fracture PFF. Among 15628 children, hospitalized in 4 neonatal units, 8 newborns were included in the study. Prematurity, low birth weight and caesarean section were suspected as PFF risk factors and Bryant’s skin traction was found as an effective option to achieve the excellent outcomes.
The paper is of interest for the research question and the low incidence of PFF. References are adequate and its readability is good. Strengths and limitations are listed by the Authors. Two points require attention:
Q1) Repetitive information in table and text should be avoided.
A1) Thanks for your comment. The repetitions were removed
Q2) Due to the very small size and subsequent limitation to perform the adequate statistical analysis, those listed as ‘risk factors’ require redefinition.
A2) Thanks for your comment. The term “Risk factors” was redefined as “predisposing factors”
Reviewer 2 Report
The authors present a retrospective study on eight neonates with perinatal femoral fractures and compare the results with those presented in the literature. Unfortunately, the paper has major limitations, which decreases the scientific value.
It is not quite clear what this study adds to the existing literature. It is rather a demographic study than an outcome study. With such small numbers, it is impossible to draw any supported conclusions.
The methods section lacks important details. E.g., how many exclusions were based on each exclusion criterion (this is important for the reader because the incidence rate is calculated based on the inclusions, while patients with too short follow-up or incomplete radiographic data were excluded).
Were >15,000 charts screened manually? How is completeness assured?
The authors describe a traction method with two legs but show a picture with only one leg in traction.
The results section is very concise, and some of the data presented do not concur with the table (e.g., birth weight, gestational age).
The duration of follow-up is described as ‘until skeletal maturation’ but the actual follow-up is not given. No follow-up investigation is described.
The incidence rates presented in the discussion are lower than those in the introduction.
The conclusions presented cannot be based on such small number.
In conclusion, although the paper is interesting, there are too many limitations as described above.
Author Response
The authors present a retrospective study on eight neonates with perinatal femoral fractures and compare the results with those presented in the literature. Unfortunately, the paper has major limitations, which decreases the scientific value.
It is not quite clear what this study adds to the existing literature. It is rather a demographic study than an outcome study. With such small numbers, it is impossible to draw any supported conclusions.
Q1) The methods section lacks important details. E.g., how many exclusions were based on each exclusion criterion (this is important for the reader because the incidence rate is calculated based on the inclusions, while patients with too short follow-up or incomplete radiographic data were excluded).
A1) Thanks for your comment. A summary figure was added in the manuscript (Figure 2).
Q2) Were >15,000 charts screened manually? How is completeness assured?
A2) Thanks for your comment. The computerized retrospective review was performed using the code “820.” (femoral fractures) on the hospitals virtual database. After the first screening, were selected the under 1 month of life patients.
Q3) The authors describe a traction method with two legs but show a picture with only one leg in traction.
A3) Thanks for your comment. The description was modified.
Q4) The results section is very concise, and some of the data presented do not concur with the table (e.g., birth weight, gestational age).
A4) Thanks for your comment. The mismatch between the text and the table was corrected.
Q5) The duration of follow-up is described as ‘until skeletal maturation’ but the actual follow-up is not given. No follow-up investigation is described.
A5) Thanks for your comment. In methods was described the orthopedic unit treatment protocol. Unfortunately, the majority of the sample (6 out of 8) was lost after 1 year from the fracture. To no provide heterogeneous and misleading data, no results after 6 months from the fracture were described.
Q6) The incidence rates presented in the discussion are lower than those in the introduction.
A6) Thanks for your comment. The mismatch between the introduction and the discussion was corrected.
Q7) The conclusions presented cannot be based on such small number.
A7) Thanks for your comment. The conclusion was modified and partially re-written. In conclusion, although the paper is interesting, there are too many limitations as described above.
Round 2
Reviewer 2 Report
The authors present a retrospective study on eight neonates with perinatal femoral fractures and compare the results with those presented in the literature. Unfortunately, the paper has major limitations, which decreases the scientific value.
It is not quite clear what this study adds to the existing literature. It is rather a demographic study than an outcome study. With such small numbers, it is impossible to draw any supported conclusions.
Q1) The methods section lacks important details. E.g., how many exclusions were based on each exclusion criterion (this is important for the reader because the incidence rate is calculated based on the inclusions, while patients with too short follow-up or incomplete radiographic data were excluded).
A1) Thanks for your comment. A summary figure was added in the manuscript (Figure 2).
The figure is helpful but the numbers are confusing. (e.g., 15628 – 15609 = 154??). Moreover, the presented incidence rate should be adjusted because some PFFs were excluded for other reasons but are true PFFs.
Q2) Were >15,000 charts screened manually? How is completeness assured?
A2) Thanks for your comment. The computerized retrospective review was performed using the code “820.” (femoral fractures) on the hospitals virtual database. After the first screening, were selected the under 1 month of life patients.
Q3) The authors describe a traction method with two legs but show a picture with only one leg in traction.
A3) Thanks for your comment. The description was modified.
Q4) The results section is very concise, and some of the data presented do not concur with the table (e.g., birth weight, gestational age).
A4) Thanks for your comment. The mismatch between the text and the table was corrected.
Q5) The duration of follow-up is described as ‘until skeletal maturation’ but the actual follow-up is not given. No follow-up investigation is described.
A5) Thanks for your comment. In methods was described the orthopedic unit treatment protocol. Unfortunately, the majority of the sample (6 out of 8) was lost after 1 year from the fracture. To no provide heterogeneous and misleading data, no results after 6 months from the fracture were described.
This should be clearly stated in the manuscript.
Q6) The incidence rates presented in the discussion are lower than those in the introduction.
A6) Thanks for your comment. The mismatch between the introduction and the discussion was corrected.
Q7) The conclusions presented cannot be based on such small number.
A7) Thanks for your comment. The conclusion was modified and partially re-written. In conclusion, although the paper is interesting, there are too many limitations as described above.
Author Response
Q1) The figure is helpful but the numbers are confusing. (e.g., 15628 – 15609 = 154??).
A1) Thank for your comment. We are sorry for the calculation error, the correct number was included in the figure 2.
Q2) Moreover, the presented incidence rate should be adjusted because some PFFs were excluded for other reasons but are true PFFs.
A2) Thank for your comment. The adjusted incidence was added in the text.
Q3) The duration of follow-up is described as ‘until skeletal maturation’ but the actual follow-up is not given. No follow-up investigation is described.
Thanks for your comment. In methods was described the orthopedic unit treatment protocol. Unfortunately, the majority of the sample (6 out of 8) was lost after 1 year from the fracture. To no provide heterogeneous and misleading data, no results after 6 months from the fracture were described.
This should be clearly stated in the manuscript.
A3) Thank for your comment. The phrase was added in results